# Geographic Differences in Phenotype and Treatment of Children with Sickle Cell Anemia from the Multinational DOVE Study

**DOI:** 10.3390/jcm8112009

**Published:** 2019-11-17

**Authors:** Baba Psalm Duniya Inusa, Raffaella Colombatti, David C. Rees, Matthew M. Heeney, Carolyn C. Hoppe, Bernhards Ogutu, Hoda M. Hassab, Chunmei Zhou, Suqin Yao, Patricia B. Brown, Lori E. Heath, Joseph A. Jakubowski, Miguel R. Abboud

**Affiliations:** 1Evelina Children’s Hospital, and Guy’s and St. Thomas’ Hospital, London SE1 7EH, UK; 2Women and Children’s Health, Faculty of Life Sciences & Medicine, King’s College, London SE1 7EH, UK; 3Clinic of Pediatric Hematology-Oncology, Azienda Ospedaliera-University of Padua, Padua 35129, Italy; rcolombatti@gmail.com; 4Department of Paediatric Haematology, King’s College London, King’s College Hospital, Denmark Hill, London SE5 8RZ, UK; David.Rees2@nhs.net; 5Dana-Farber/Boston Children’s Cancer and Blood Disorders Center, Boston, MA 02115, USA; Matthew.Heeney@childrens.harvard.edu; 6UCSF Benioff Children’s Hospital Oakland, Oakland, CA 94609, USA; Choppe@mail.cho.org; 7U.S. Army Medical Research Unit-Kenya, Centre for Clinical Research, Kenya Medical Research Institute, Kisumu 00200, Kenya; ogutu6@gmail.com; 8Pediatric Department and Clinical Research Center, Faculty of Medicine, Alexandria University, Alexandria 21500, Egypt; drhoda@doctor.com; 9Eli Lilly and Company, Indianapolis, IN 46285, USA; zhou_chunmei@lilly.com (C.Z.); yao_suqin_susan@lilly.com (S.Y.); Pat6271@sbcglobal.net (P.B.B.); lorieheath@live.com (L.E.H.); jayacjay@aol.com (J.A.J.); 10Department of Pediatrics and Adolescent Medicine, American University of Beirut, Beirut 11-0236, Lebanon; abboudm@aub.edu.lb

**Keywords:** sickle cell disease, geographic, phenotypic pattern, vaso-occlusive crises (voc), global

## Abstract

Background: DOVE (Determining Effects of Platelet Inhibition on Vaso-Occlusive Events) was a Phase 3, randomized, double-blind, placebo-controlled study conducted in children with sickle cell anemia at 51 sites in 13 countries across four continents. Procedure: Data from DOVE were assessed for regional differences in subject phenotype and treatment. Demographics, baseline clinical and laboratory data, hydroxyurea (HU) use, vaso-occlusive crisis (VOCs; composite endpoint of painful crisis or acute chest syndrome (ACS)), serious adverse events (SAEs), hospitalization, and treatments were compared across the Americas, Europe, North Africa/Middle East, and Sub-Saharan Africa (SSA). Results: Race, body mass index, and blood pressures differed by region. Pre-enrollment VOCs were highest in the Americas. For subjects not on HU, baseline hemoglobin was lowest in SSA; reticulocyte count was lowest in the Americas. Within SSA, Kenya subjects presented higher baseline hemolysis. Painful crisis was the most common SAE, followed by ACS in the Americas and infections in other regions. VOC rate and percentage of VOC hospitalizations were highest in Europe. Regardless of region, most VOCs were treated with analgesics; approximately half were treated with intravenous fluids. The proportion of VOC-related transfusions was greatest in Europe. Lengths of hospital stay were similar across regions. Conclusions: Overall differences in SAEs and hospitalization for VOCs may be due to cultural diversities, resource utilization, disease severity, or a combination of factors. These data are of importance for the planning of future trials in SCA in a multinational setting.

## 1. Introduction

Sickle cell anemia (SCA) is a monogenic blood disorder that affects more than 300,000 newborns worldwide each year [1]. The majority of affected births occur in the developing world, with an estimated 237,000 infants with SCA born annually in Sub-Saharan Africa (SSA) [1] SCA manifests phenotypic variability, ranging from early-onset debilitating pain in early childhood, stroke, and severe infections, to asymptomatic cases [2,3]. A substantial proportion of these differences can be accounted for by genetic factors, such as fetal hemoglobin level or alpha-thalassemia co-inheritance [4]. However, phenotypic variability may also be affected by factors such as climate, access to or quality of medical care, and adherence to prophylactic treatment [5,6,7].

DOVE (Determining Effects of Platelet Inhibition on Vaso-Occlusive Events) was a Phase 3, double-blind, placebo-controlled, parallel-groups, multinational study designed to assess the efficacy and safety of prasugrel, a P2Y12 adenosine diphosphate receptor antagonist, for reduction of vaso-occlusive crises (VOCs) in children with SCA. DOVE was one of the largest, randomized, placebo-controlled studies of children with SCA, conducted at 51 sites in 13 countries across four continents: the Americas, Europe, North Africa/Middle East, and SSA (Figure 1). In each country, the study was approved by national regulatory authorities and by local ethics committees, institutional review boards, or both, and abided by local regulations. The design [8] and outcomes [9] of DOVE have been described previously.

Because DOVE recruited subjects with SCA only (homozygous hemoglobin S (HbSS) and hemoglobin Sβ (0) thalassemia (HbSβ0))—genotypes with similar features [10]—geographic differences in pre-study and intra-study disease may be ascribed to other factors. Data from DOVE were analyzed for regional differences in subject demographics and baseline clinical characteristics, baseline laboratory values, environmental factors, such as malaria infection, and therapeutic interventions, such as hydroxyurea (HU) utilization. The study demonstrated that large-scale multinational trials are possible in SCA regardless of regional differences.

## 2. Methods

### 2.1. Study Population

A total of 341 children, aged 2 to <18 years, with SCA, who had at least two documented VOCs in the previous year, were randomized. VOC was the composite endpoint of painful crisis or acute chest syndrome (ACS, Beijing, China). HU therapy was allowed if subjects were on a stable dose for at least 60 days prior to randomization. Subjects aged 2 to 16 years with abnormal or conditional transcranial Doppler within the past year were excluded. Other selection criteria have been described previously [8,9].

### 2.2. Randomization and Masking

Subjects were randomized 1:1 in a double-blind fashion to prasugrel (*n* = 171) or placebo (*n* = 170), with treatment assignment balanced by HU use, country, and age group (cohorts: 2 to <6 years, 6 to <12 years, and 12 to <18 years) [8,9].

### 2.3. Statistical Analyses

Analyses for baseline characteristics and efficacy endpoints were performed with data from the intent-to-treat population; safety analyses were performed with data from subjects who received at least one dose of the study drug. Both arms (prasugrel and placebo) were pooled in all analyses for this study. VOC event rates were compared among geographic regions using the Andersen–Gill model. A robust variance estimator was used with geographic region, HU use, and age group included as factors in the model. For all other statistical comparisons among geographic regions, a Fisher’s exact test was used for categorical variables; an ANOVA model, adjusting for age group and HU use, was used for continuous variables.

## 3. Results

### 3.1. Demographics of the Study Population

The highest enrolling regions were SSA (N = 148) and North Africa/Middle East (N = 110), followed by the Americas (N = 57) and Europe (N = 26). The majority of enrolled subjects had the HbSS genotype, accounting for 77.3% of subjects from North Africa/Middle East, 84.6% from Europe, 93.0% from the Americas, and 100% from SSA. As shown in Table 1, the highest proportion of 2- to <6-year-olds enrolled in Europe, 6- to <12-year-olds enrolled in SSA, and 12- to <18-year-olds enrolled in North Africa/Middle East. Racial groups differed by region, with the percentage of subjects self-reporting as “white” ranging from 0% (SSA) to 100% (North Africa/Middle East). Approximately 50% of subjects from each region were female.

### 3.2. Medical History and Baseline Clinical Characteristics

Body mass index (BMI) was lower in subjects from SSA and Europe than other regions (Table 2). Subjects from SSA had the lowest systolic and diastolic blood pressures (Table 2). Among 6 to <12-year-olds, subjects from SSA had the lowest systolic blood pressure. Among 12 to <18-year-olds, subjects from SSA had the lowest diastolic blood pressure (Appendix A).

HU use at baseline differed by region: 6.8% of subjects in SSA (0% in Ghana), 42.3% in Europe, 72.7% in North Africa/Middle East, but 91.2% in the Americas (Table 2). In the Americas, mean number of VOCs in the year prior to enrollment was higher than other regions. Pre-study VOCs were primarily due to painful crisis rather than ACS; approximately 22% of the subjects reported history of ACS, half of whom were enrolled in the Americas (38 of 76 subjects) (Table 2).

In DOVE, 31 subjects reported a history of splenic sequestration, with the highest proportions enrolled in the Americas (38.7% (12 of 31 subjects)) and North Africa/Middle East (32.3% (ten of 31 subjects)). Similarly, the majority of subjects with a history of splenectomy were from the Americas (34.4% (11 of 32 subjects)) or North Africa/Middle East (62.5% (20 of 32 subjects)) (Table 2). However, North Africa/Middle East was the only region with a greater proportion of subjects having undergone splenectomy compared to the proportion reporting a history of splenic sequestration (Table 2). Further examination showed that six of the 20 subjects from this region with previous splenectomy had the HbSβ0 genotype.

### 3.3. Baseline Laboratory Values 

Subjects Not on HU Therapy: Reticulocyte count was highest in SSA and lowest in the Americas whereas hemoglobin level was lowest in SSA and highest in the Americas (Table 3). Baseline platelet counts were highest in subjects from SSA compared to other regions (Table 3). In contrast, platelet volume was highest in the Americas and Europe but lowest in SSA. There were no regional differences observed for total bilirubin, lactic dehydrogenase, mean cell volume, or leukocyte count (Table 3). Further analysis of subjects from SSA showed that total bilirubin, platelet count, reticulocyte count, and leukocyte count were significantly higher in subjects from East Africa (Kenya) compared to West Africa (Ghana) (Appendix A).

Subjects on HU Therapy: Few regional differences in baseline laboratory values were observed. Mean baseline platelet volume was lowest in SSA compared to the other regions (8.3 vs. ≥9.0 fL; *p* = 0.002), whereas mean reticulocyte count was higher in SSA (275.9 vs. ≤189.7 × 10^9^/L; *p* = 0.027). Leukocyte count was highest in SSA and North Africa/Middle East (11.3 and 11.0 × 10^9^/L, respectively) and lowest in the Americas and Europe (8.5 and 7.8 × 10^9^/L, respectively) (*p* = 0.006).

### 3.4. Vaso-Occlusive Crises during the Study

During DOVE, 818 VOCs were documented and experienced by 238 of the 341 subjects enrolled, with the majority of VOCs reported as painful crisis (96%) and very few reported as ACS (4%). The overall rate of VOCs (events per patient-year) differed by region (*p* = 0.003): 3.2 in Europe, 3.0 in the Americas, 2.6 in SSA, and 2.0 in North Africa/Middle East. The highest proportion of subjects who experienced ACS occurred in Europe (30.8%) compared to other regions (SSA, 4.1%; North Africa/Middle East, 6.4%; the Americas, 15.8%).

The percentage of VOCs associated with hospitalization during the study was highest in Europe, followed by North Africa/Middle East, the Americas, and SSA (Figure 2). However, the mean length of hospital stay per VOC was similar across regions (5.3–6.2 days, *p* = 0.32). In SSA, the majority of VOCs were managed as outpatient hospital visits whereas subjects from other regions were more likely to be admitted to the hospital (Table 4). Telephone consultation for VOCs was used mainly in the Americas and North Africa/Middle East, whereas home support visits for VOCs (including follow-up telephone calls for painful crisis) were utilized most often in Europe (Table 4).

### 3.5. Serious Adverse Events during the Study

The percentage of subjects with at least one serious adverse event (SAE) during the study was 44.6% in SSA, 55% in North Africa/Middle East, 63.2% in the Americas, and 84.6% in Europe. The most frequent SAE in all regions was painful crisis (SSA, 27%; North Africa/Middle East, 44%; the Americas, 57.9%; Europe, 69.2%). The second most frequent SAE was ACS in the Americas (12.3%), malaria in SSA (14.2%), acute tonsillitis in North Africa/Middle East (2.8%), and viral respiratory tract infections in Europe, the majority of which were respiratory (23.1%).

Malaria infection was reported only for SSA. Further analysis showed that 40.4% of subjects from Ghana and 48.4% of subjects from Kenya reported malaria during the study but only 17.5% and 12.1%, respectively, were designated as SAEs. All cases reported as SAEs were associated with hospitalization.

### 3.6. Therapeutic Interventions during the Study

Regardless of region, almost all VOCs were treated with analgesics and approximately half were treated with intravenous (IV) fluids. In contrast, the proportion of VOC-related transfusions was greater in Europe and North Africa/Middle East than other regions (Table 4).

Whole blood was used for transfusion primarily in SSA and to a much lesser extent in North Africa/Middle East, but was not used in the Americas or Europe. All regions used packed red blood cells; SSA was the only region where leukoreduced blood components were not used. Transfusion with fresh frozen plasma or platelets was rare and was utilized for only one subject each in North Africa/Middle East.

## 4. Discussion

DOVE was one of the largest Phase 3 studies of children with SCA and examined subject differences in phenotype, relevant clinical measurements, and treatment patterns across four major continents. The majority of treatment-emergent VOCs during the study were reported as painful crisis (96%) compared to ACS (4%).

Subjects from the Americas had the highest mean number of VOCs prior to enrollment and the second highest VOC rate during the study, despite the fact that majority were on HU therapy at the time of randomization. HU improves disease outcomes in subjects with SCA by increasing fetal hemoglobin and reducing the number of VOCs [11]. It is possible that children enrolled in the Americas may represent subjects with more severe disease with breakthrough pain despite HU use. Of interest, Kanter et al. [12] found that use of opioid, compared to non-opioid, analgesics in DOVE was higher in the Americas than in any other region. It is perhaps surprising that acute pain and ACS were least reported in SSA even though utilization of HU was limited. The potential reasons may include regional differences in subject selection; perhaps regions enrolling more limited numbers of subjects included individuals with more severe phenotypes, who were more willing to participate in a “new drug” trial, compared to SSA which had a larger subject pool from which to recruit participants. It is also likely that differences exist for pain perception, therapeutic approaches for pain, or external environmental factors (such as climate [7]), and account for the disparity. There are potential cultural differences in terms of family support through the network of the extended family that is likely offer supportive advice and assist in child care. In high income settings where parents may be first generation immigrants and lack the necessary support systems will rely on social and health systems as buffers.

Painful crisis and ACS are the most common causes of hospitalization among children with SCA. [13,14] Study subjects from Europe had the highest percentage of VOC-associated hospitalizations whereas SSA had the lowest; however, mean length of hospital stay per VOC did not differ by region. The regional findings in VOC-associated hospitalizations did not appear to correspond with laboratory indices of anemia, hemolysis, or inflammation. Regional differences in organization of the health care system may account, at least in part, for these findings; for example, the availability of free health care in Europe compared to other regions could explain the higher rates of hospital admissions in this region.

Children with SCA are at increased risk for infection. [15] In DOVE, infections were the second most frequent SAE in SSA, North Africa/Middle East, and Europe, with the types of infections varying by region. Malaria occurred only in SSA. An inverse relationship between malaria parasitemia and hemoglobin level has been identified [16] and may explain why subjects from SSA had the lowest concentrations of hemoglobin.

Platelet counts were highest for subjects from SSA, which was associated with an expected lower mean platelet volume than other areas. This inverse relationship was found in all geographic regions of the DOVE study, and has been documented in healthy populations [17] Segal and Moliterno [18] found that subjects of African descent had higher platelet counts than other ethnic groups that were not the result of environmental factors and were possibly genetic and/or hormonally driven. Celik et al. [19] reported that platelet volume was higher in subjects with cerebrovascular events and increased with incidence of VOCs, suggesting that higher mean platelet volumes may be an early predictor of future cerebrovascular events. Although additional research would be required, it is interesting to note that subjects from SSA, who had the lowest mean platelet volume at baseline, also had low VOC rates during the study. Surprisingly, children in East Africa (Kenya) presented higher baseline indices of hemolysis compared to those in West Africa (Ghana); these data coincide with regional differences in malaria burden and warrant further investigations into the role of environmental or even genetic influences on laboratory values.

Acute splenic sequestration is a life-threatening complication of SCA that is associated with high mortality rates and frequent recurrence in subjects surviving the first attack [20], especially in low-income countries. Although splenectomy is often indicated for splenic sequestration [20], there is little data from clinical trials to show that the procedure improves survival and decreases morbidity in subjects with SCA [21]. In DOVE, a proportion of subjects from each region had splenic sequestration prior to enrollment; only the North Africa/Middle East region reported more subjects with a history of splenectomy compared to splenic sequestration. Literature suggests that subjects may also undergo splenectomy for hypersplenism, splenic abscess, massive splenic infarction, or increased requirement for packed red blood cell transfusion [20,21,22], which may explain the findings in North Africa/Middle East. It would be of interest to study how age and indication for splenectomy vary across region and SCA genotype in future clinical trials developed at a global level.

Even though subjects from SSA had the lowest BMI, comparing the mean age and BMI for each region to the Centers for Disease Control and Prevention charts [23] for boys and girls indicates that DOVE participants were of healthy/normal BMI.

Patients with SCA have lower blood pressures than published norms [24,25,26]. In DOVE, subjects from SSA had lower blood pressures than subjects from other regions. Using charts published by Banker et al. [27] which assess blood pressure as a function of height, systolic and diastolic blood pressures of subjects enrolled in DOVE fell within the normal range except that systolic blood pressures of children from the Americas approached the pre-hypertensive range. This finding may be related to cultural differences in diet and exercise.

Analgesics and IV fluids are commonly used to treat VOCs in subjects with SCA [28] and utilization of such supportive measures was similar across geographic regions in DOVE. During DOVE, subjects from Europe were most likely to receive a blood transfusion whereas subjects from SSA were least likely. Regional differences in transfusion criteria may explain these findings and may stem, at least in part, from regional differences in availability of blood products [29] or transfusion guidelines.

## 5. Conclusions

In summary, the subjects of DOVE differed across regions with regard to baseline laboratory values, rate of VOCs during the study, and VOC-associated hospitalizations and transfusions. However, the management of VOCs with analgesics and IV fluids was similar across the regions. Regional differences in SCA clinical presentation or intra-study treatment during the DOVE study may have been influenced by regional differences in culture, utilization of resources, disease severity, climate, or a combination of factors. While the findings of the DOVE study reaffirm that pain is the most common complication of SCA, the observed regional differences in the study call for more detailed research to understand the impact of environmental determinants of pain and cultural approaches to therapeutic intervention. Several investigators have concerns about the feasibility of conducting trials in SCD outside Europe and North America. This study is proof that clinical trials are feasible in the continent with the largest burden of disease and stand to benefit the most from the success in new therapies. Gone are the days when they are just presented therapies as a given, it will make it possible to test and try the drugs within the local context. As new agents are identified that may impact the disease process in SCD it is imperative that large scale trials be conducted. These must, of necessity, involve sites in the Middle East and SSA in order to accrue sufficient patients. Our data clearly demonstrate that this is possible and that, regardless of regional differences in management, the disease process in SCA seems remarkably similar.

## Figures and Tables

**Figure 1 jcm-08-02009-f001:**
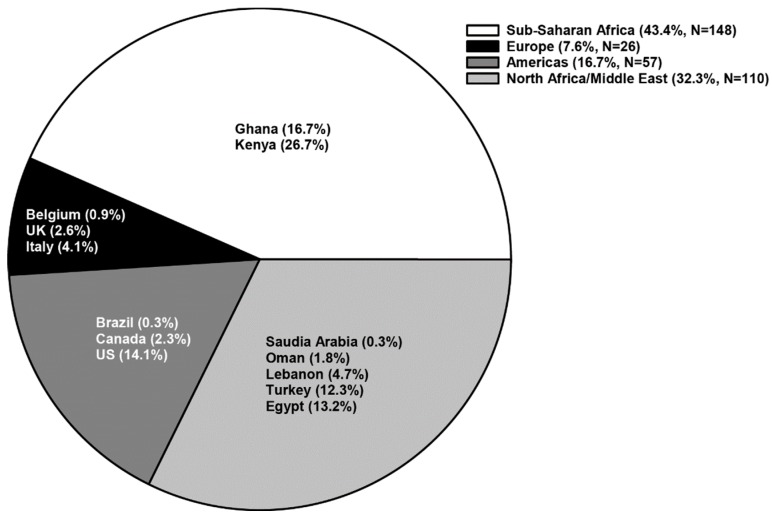
Percent enrollment per region: N, number of randomized subjects; UK, United Kingdom; US, United States.

**Figure 2 jcm-08-02009-f002:**
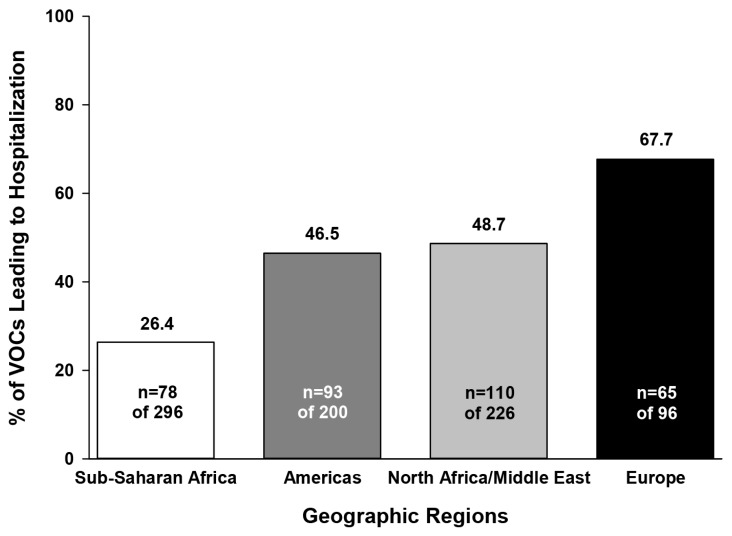
Percentage of vaso-occlusive crises associated with hospitalization during the DOVE study. Note: locations could have included inpatient hospitalizations, outpatient hospitalizations, and emergency room visits: n, number of events in the specified category; VOCs, vaso-occlusive crises.

**Table 1 jcm-08-02009-t001:** Key demographics of subjects enrolled in the DOVE study in the different geographic regions.

Parameter Assessed	Geographic Regions ^a^				Total (N = 341)	*p*-Value
	SSA (N = 148)	Americas (N = 57)	North Africa/Middle East (N = 110)	Europe (N = 26)		
Mean Age in Years (SD)	9.7 (3.9)	11.1 (4.2)	12.0 (4.6)	8.7 (4.3)	10.6 (4.3)	0.020
Age Group (*n* (%))						<0.001
2 to <6 years	32 (21.6)	9 (15.8)	17 (15.5)	9 (34.6)	67 (19.6)	
6 to <12 years	72 (48.6)	22 (38.6)	29 (26.4)	9 (34.6)	132 (38.7)	
12 to <18 years	44 (29.7)	26 (45.6)	64 (58.2)	8 (30.8)	142 (41.6)	
Gender (*n* (%))						0.733
Female	75 (50.7)	32 (56.1)	52 (47.3)	14 (53.8)	173 (50.7)	
Race (*n* (%)) ^b^						<0.001
White	0 (0.0)	1 (1.8)	110 (100.0)	5 (19.2)	116 (34.1)	
Black or African American	148 (100.0)	54 (96.4)	0 (0.0)	20 (76.9)	222 (65.3)	
Multiple	0 (0.0)	1 (1.8)	0 (0.0)	1 (3.8)	2 (0.6)	

Abbreviations: N = number of subjects in the specified category; N = number of randomized subjects; SD = standard deviation; SSA = Sub-Saharan Africa. ^(**a**)^ The SSA subgroup includes Ghana and Kenya; the Americas subgroup includes Brazil, the United States, and Canada; the North Africa/Middle East subgroup includes Saudi Arabia, Oman, Egypt, Lebanon, and Turkey; the Europe subgroup includes Belgium, Italy, and the United Kingdom (Heeney et al.) [9]. ^(**b**)^ Refers to self-reported race.

**Table 2 jcm-08-02009-t002:** Key baseline characteristics that significantly differed among the geographic regions.

Parameter Assessed	Geographic Regions ^a^				Total (N = 341)	*p*-Value
	SSA (N = 148)	Americas (N = 57)	North Africa/Middle East (N = 110)	Europe (N = 26)		
HU Use at Baseline, *n* (%)	10 (6.8)	52 (91.2)	80 (72.7)	11 (42.3)	153 (44.9)	<0.001
BMI (kg/m^2^), mean (SD)	15.3 (2.0)	18.1 (3.5)	18.3 (3.6)	16.5 (2.9)	16.8 (3.2)	<0.001
Blood Pressure, Systolic (mmHg), mean (SD)	99.0 (9.7)	108.0 (10.7)	105.4 (11.7)	105.6 (12.0)	103.0 (11.3)	0.004
Blood Pressure, Diastolic (mmHg), mean (SD)	58.3 (6.9)	61.1 (7.3)	62.9 (9.4)	60.4 (8.5)	60.4 (8.2)	0.003
No. of VOCs in Prior Year, mean (SD)	3.4 (1.7)	5.8 (13.5)	3.2 (1.8)	3.2 (1.6)	3.7 (5.8)	0.041
Diagnosed with ACS Pre-enrollment, *n* (%)	9 (6.1)	38 (66.7)	20 (18.2)	9 (34.6)	76 (22.3)	<0.001
Diagnosed with Splenic Sequestration Pre-enrollment, *n* (%)	7 (4.7)	12 (21.1)	10 (9.1)	2 (7.7)	31 (9.1)	0.007
Undergone Splenectomy Pre-enrollment, *n* (%)	0 (0.0)	11 (19.3)	20 (18.2)	1 (3.8)	32 (9.4)	<0.001

Abbreviations: ACS = acute chest syndrome; BMI = body mass index; HU = hydroxyurea; n = number of subjects in the specified category; N = number of randomized subjects; No. = number; SD = standard deviation; SSA = Sub-Saharan Africa; VOCs = vaso-occlusive crises. ^(**a**)^ The SSA subgroup includes Ghana and Kenya; the Americas subgroup includes Brazil, the United States, and Canada; the North Africa/Middle East subgroup includes Saudi Arabia, Oman, Egypt, Lebanon, and Turkey; the Europe subgroup includes Belgium, Italy, and the United Kingdom (Heeney et al.) [9].

**Table 3 jcm-08-02009-t003:** Key baseline laboratory values by region for subjects not on HU.

Parameter Assessed (Mean (SD))	Geographic Regions ^a^				*p*-Value
	SSA (*n* = 131)	Americas (*n* = 4)	North Africa/Middle East (*n* = 28)	Europe (*n* = 15)	
Total Bilirubin (µM/L)	39.2 (27.2)	24.8 (7.8)	41.6 (29.6)	33.8 (8.7)	0.729
Hemoglobin (mg/L)	47 (0.6)	62 (0.8)	53 (1.0)	52 (0.5)	<0.001
Lactic Dehydrogenase (units/L)	569.6 (159.9)	554.7 (311.3)	495.3 (216.9)	519.4 (140.4)	0.272
Mean Cell Volume (fL)	84.7 (10.2)	89.3 (10.8)	85.4 (11.6)	85.5 (5.9)	0.812
Platelet Volume (fL)	9.0 (0.7)	10.1 (1.3)	9.9 (1.4)	10.1 (1.4)	<0.001
Platelet Count (×10^9^/L)	440.3 (141.7)	330.3 (167.8)	332.2 (176.3)	331.7 (91.5)	<0.001
Reticulocyte Count (×10^9^/L)	318.6 (117.8)	214.8 (75.4)	238.9 (107.7)	327.8 (101.1)	0.004
Leukocyte Count (×10^9^/L)	15.0 (4.4)	11.6 (7.3)	12.7 (6.0)	14.0 (5.8)	0.061

Abbreviations: HU = hydroxyurea; *n* = number of subjects with at least 1 parameter measured at baseline; SD = standard deviation; SSA = Sub-Saharan Africa. ^(**a**)^ The SSA subgroup includes Ghana and Kenya; the Americas subgroup includes Brazil, the United States, and Canada; the North Africa/Middle East subgroup includes Saudi Arabia, Oman, Egypt, Lebanon, and Turkey; the Europe subgroup includes Belgium, Italy, and the United Kingdom (Heeney et al.) [9].

**Table 4 jcm-08-02009-t004:** Management of VOCs during the DOVE study.

Parameter Assessed (*n* (%))	Geographic Regions ^a^				
	SSA (N = 148)	Americas (N = 57)	North Africa/Middle East (N = 110)	EuropeZ (N = 26)	Total (N = 341)
Total Number of VOCs	296	200	226	96	818
Location of Medical Intervention ^b^					
VOCs managed by outpatient hospital visit	201 (67.9)	10 (5.0)	44 (19.5)	6 (6.3)	261 (31.9)
VOCs managed by inpatient hospital visit	71 (24.0)	89 (44.5)	75 (33.2)	53 (55.2)	288 (35.2)
VOCs managed by home support visits ^c^	10 (3.4)	18 (9.5)	1 (0.5)	16 (18.4)	45 (5.7)
VOCs managed by telephone consultation ^c^	5 (1.7)	52 (27.4)	46 (21.1)	5 (5.7)	108 (13.8)
VOCs managed by emergency room visit	9 (3.0)	24 (12.0)	58 (25.7)	16 (16.7)	107 (13.1)
Types of Medical Intervention					
VOCs requiring analgesics	295 (99.7)	199 (99.5)	224 (99.1)	96 (100.0)	814 (99.5)
VOCs requiring IV fluids	136 (45.9)	113 (56.5)	150 (66.4)	46 (47.9)	445 (54.4)
VOCs requiring transfusion	19 (6.4)	20 (10.0)	42 (18.6)	18 (18.8)	99 (12.1)

Abbreviations: ACS = acute chest syndrome; IV = intravenous; n = number of events in the specified category; N = number of randomized subjects; SSA = Sub-Saharan Africa; VOCs = vaso-occlusive crises. ^(**a**)^ The SSA subgroup includes Ghana and Kenya; the Americas subgroup includes Brazil, the United States, and Canada; the North Africa/Middle East subgroup includes Saudi Arabia, Oman, Egypt, Lebanon, and Turkey; the Europe subgroup includes Belgium, Italy, and the United Kingdom (Heeney et al.) [9]. ^(**b**)^ Professional health care office/clinic visits and urgent care visits are not included in the table since all regions utilized them for ≤2% of VOCs. ^(**c**)^ Home support visits and telephone consultations were available for episodes of painful crisis only, not ACS.

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
