# Peer review of "Geographic Differences in Phenotype and Treatment of Children with Sickle Cell Anemia from the Multinational DOVE Study"

_jcm, 2019, doi:10.3390/jcm8112009_

Round 1

Reviewer 1 Report

The study from Inusa et al describes regional differences among patients enrolled in the DOVE clinical study,  one of the best designed and conducted clinical trial in sickle cell disease. The authors take advantage of the negative results of the clinical study to combine the data from placebo and Prasugrel treated patients. The paper is generally well written and the data are clearly presented.

Unluckily the DOVE study was not designed to properly analyze regional related sickle cell phenotypes and big differences in the number of analyzed patients, among the countries, particularly related to patients not on HU, represent a weakness, possibly leading to wrong conclusions. Clusterization of patients with different ages would improve the outcomes, but probably the small number of patients in some regional groups did not allowed it. Despite these limits the paper points out  that the recruitment of patients in multinational studies shows differences linked to geographical localization and these should be considered to limit statistical pitfalls.

Author Response

1. Unluckily the DOVE study was not designed to properly analyze regional related sickle cell phenotypes and big differences in the number of analyzed patients,

We agree with this comments, this was secondary analysis to show the importance of comparing clinical data and patient outcomes for sickle cell disease beyond high income settings. This nevertheless raises possible future research themes. It also offers the opportunity to expand clinical trials to patients in sub-Saharan Africa bearing in mind some variation in management of patients. 

Reviewer 2 Report

This is an interesting descriptive study looking at a broad number of patients with sickle cell anemia across a diverse geographic region.  I would suggest some edits that would improve the impact of the paper:

-nowhere is there mention of sickle cell haplotype, which may vary by region and may affect disease severity in the different populations

-the authors mention regional differences in treatment, and cultural issues, which almost certainly effect these results, but do not elaborate sufficiently on what these may be

Author Response

nowhere is there mention of sickle cell haplotype, which may vary by region and may affect disease severity in the different populations.

We did not investigate the different haplotypes in this particular trial. We acknowledge the possibility of different proportions of different haplotypes as described in the literature, Senegal and Indo-arab which express higher level of fetal hemoglobin than  the Bantu and Benin Haplotypes see Rees DC1, Williams TN, Gladwin MT in Lancet. 2010 Dec 11;376(9757):2018-31. doi: 10.1016/S0140-6736(10)61029-X. Epub 2010 Dec 3.

 it was later at the end of the the trial the sub-optimal platelet inhibition may in fact have affected the negative outcome of the trial as reported. Thromb Haemost. 2017 Feb 28;117(3):580-588. doi: 10.1160/TH16-09-0731. Epub 2016 Dec 8.

2. the authors mention regional differences in treatment, and cultural issues, which almost certainly effect these results, but do not elaborate sufficiently on what these may be. 

- we have highlighted the possible factors th diferences in page 9 of paper

The potential reasons may include regional differences in subject selection; perhaps regions enrolling more limited numbers of subjects included individuals with more severe phenotypes, who were more willing to participate in a “new drug” trial, compared to SSA which had a larger subject pool from which to recruit participants. It is also likely that differences exist for pain perception, therapeutic approaches for pain, or external environmental factors (such as climate7), and account for the disparity.